# Scattering GCN: Overcoming Oversmoothness in Graph Convolutional Networks

**Yimeng Min**[*]
Mila – Quebec AI Institute
Montreal, QC, Canada
`minyimen@mila.quebec`

**Frederik Wenkel**[*]
Dept. of Math. and Stat.
Université de Montréal
Mila – Quebec AI Institute
Montreal, QC, Canada
`frederik.wenkel@umontreal.ca`

**Guy Wolf**
Dept. of Math. and Stat.
Université de Montréal
Mila – Quebec AI Institute
Montreal, QC, Canada
`guy.wolf@umontreal.ca`

## Abstract

Graph convolutional networks (GCNs) have shown promising results in processing graph data by extracting structure-aware features. This gave rise to extensive work in geometric deep learning, focusing on designing network architectures that ensure neuron activations conform to regularity patterns within the input graph. However, in most cases the graph structure is only accounted for by considering the similarity of activations between adjacent nodes, which limits the capabilities of such methods to discriminate between nodes in a graph. Here, we propose to augment conventional GCNs with geometric scattering transforms and residual convolutions. The former enables band-pass filtering of graph signals, thus alleviating the so-called oversmoothing often encountered in GCNs, while the latter is introduced to clear the resulting features of high-frequency noise. We establish the advantages of the presented Scattering GCN with both theoretical results establishing the complementary benefits of scattering and GCN features, as well as experimental results showing the benefits of our method compared to leading graph neural networks for semi-supervised node classification, including the recently proposed GAT network that typically alleviates oversmoothing using graph attention mechanisms.

## 1 Introduction

Deep learning approaches are at the forefront of modern machine learning. While they are effective in a multitude of applications, their most impressive results are typically achieved when processing data with inherent structure that can be used to inform the network architecture or the neuron connectivity design. For example, image processing tasks gave rise to convolutional neural networks that rely on spatial organization of pixels, while time series analysis gave rise to recurrent neural networks that leverage temporal organization in their information processing via feedback loops and memory mechanisms. The success of neural networks in such applications, traditionally associated with signal processing, has motivated the emergence of geometric deep learning, with the goal of generalizing the design of structure-aware network architectures from Euclidean spatiotemporal structures to a wide range of non-Euclidean geometries that often underlie modern data.

Geometric deep learning approaches typically use graphs as a model for data geometries, either by constructing them from input data (e.g., via similarity kernels) or directly given as quantified interactions between data points [1]. Using such models, recent works have shown that graph neural networks (GNNs) perform well in multiple application fields, including biology, chemistry and social networks [2–4]. It should be noted that most GNNs consider each graph together with given

---

[*]Equal contribution; order determined alphabetically

node features, as a generalization of images or audio signals, and thus aim to compute whole-graph representations. These in turn, can be applied to graph classification, for example when each graph represents the molecular structure of proteins or enzymes classified by their chemical properties [5–7].

On the other hand, methods such as graph convolutional networks (GCNs) presented by [4] consider node-level tasks and in particular node classification. As explained in [4], such tasks are often considered in the context of semi-supervised learning, as typically only a small portion of nodes of the graph possesses labels. In these settings, the entire dataset is considered as one graph and the network is tasked with learning node representations that infer information from node features as well as the graph structure. However, most state-of-the-art approaches for incorporating graph structure information in neural network operations aim to enforce similarity between representations of adjacent (or neighboring) nodes, which essentially implements local smoothing of neuron activations over the graph [8]. While such smoothing operations may be sufficiently effective in whole-graph settings, they often cause degradation of results in node processing tasks due to oversmoothing [8, 9], as nodes become indistinguishable with deeper and increasingly complex network architectures. Graph attention networks [10] have shown promising results in overcoming such limitations by introducing adaptive weights for graph smoothing via message passing operations, using attention mechanisms computed from node features and masked by graph edges. However, these networks still essentially rely on enforcing similarity (albeit adaptive) between neighboring nodes, while also requiring more intricate training as their attention mechanism requires gradient computations driven not only by graph nodes, but also by graph edges. We refer the reader to the supplement for further discussion of related work and recent advances in node processing with GNNs.

In this paper, we propose a new approach for node-level processing in GNNs by introducing neural pathways that encode higher-order forms of regularity in graphs. Our construction is inspired by recently proposed geometric scattering networks [11–13], which have proven effective for whole-graph representation and classification. These networks generalize the Euclidean scattering transform, which was originally presented by [14] as a mathematical model for convolutional neural networks. In graph settings, the scattering construction leverages deep cascades of graph wavelets [15, 16] and pointwise nonlinearities to capture multiple modes of variation from node features or labels. Using the terminology of graph signal processing, these can be considered as generalized band-pass filtering operations, while GCNs (and many other GNNs) can be considered as relying on low-pass filters only. Our approach combines together the merits of GCNs on node-level tasks with those of scattering networks known from whole-graph tasks, by enabling learned node-level features to encode geometric information beyond smoothed activation signals, thus alleviating oversmoothing concerns often raised in GCN approaches. We discuss the benefits of our approach and demonstrate its advantages over GCNs and other popular graph processing approaches for semi-supervised node classification, including significant improvements on the DBLP graph dataset from [17].

**Notations:** We denote matrices and vectors with bold letters with uppercase letters representing matrices and lowercase letters representing vectors. In particular, $\boldsymbol{I}_n \in \mathbb{R}^{n \times n}$ is used for the identity matrix and $\boldsymbol{1}_n \in \mathbb{R}^n$ denotes the vector with ones in every component. We write $\langle ., . \rangle$ for the standard scalar product in $\mathbb{R}^n$. We will interchangeably consider functions of graph nodes as vectors indexed by the nodes, implicitly assuming a correspondence between a node and a specific index. This carries over to matrices, where we relate nodes to column or row indices. We further use the abbreviation $[n] := \{1, \ldots, n\}$ where $n \in \mathbb{N}$ and write $\mathbb{N}_0 := \mathbb{N} \cup \{0\}$.

## 2 Graph Signal Processing

Let $G = (V, E, w)$ be a weighted graph with $V := \{v_1, \ldots, v_n\}$ the set of nodes, $E \subset \{\{v_i, v_j\} \in V \times V, i \neq j\}$ the set of (undirected) edges and $w : E \to (0, \infty)$ assigning (positive) edge weights to the graph edges. We note that $w$ can equivalently be considered as a function of $V \times V$, where we set the weights of non-adjacent node pairs to zero. We define a *graph signal* as a function $x : V \to \mathbb{R}$ on the nodes of $G$ and aggregate them in a signal vector $\boldsymbol{x} \in \mathbb{R}^n$ with the $i^{th}$ entry being $x(v_i)$.

We define the (combinatorial) *graph Laplacian* matrix $\boldsymbol{L} := \boldsymbol{D} - \boldsymbol{W}$, where $\boldsymbol{W} \in \mathbb{R}^{n \times n}$ is the *weighted adjacency matrix* of the graph $G$ given by

$$\boldsymbol{W}[v_i, v_j] := \begin{cases} w(v_i, v_j) & \text{if } \{v_i, v_j\} \in E \\ 0 & \text{otherwise} \end{cases},$$

and $\boldsymbol{D} \in \mathbb{R}^{n \times n}$ is the *degree matrix* of $G$ defined by $\boldsymbol{D} := \mathrm{diag}(d_1, \ldots, d_n)$ with $d_i := \deg(v_i) := \sum_{j=1}^{n} \boldsymbol{W}[v_i, v_j]$ being the *degree* of the node $v_i$. In practice, we work with the (symmetric) *normalized Laplacian* matrix $\mathcal{L} := \boldsymbol{D}^{-1/2} L \boldsymbol{D}^{-1/2} = \boldsymbol{I}_n - \boldsymbol{D}^{-1/2} \boldsymbol{W} \boldsymbol{D}^{-1/2}$. It can be verified that $\mathcal{L}$ is symmetric and positive semi-definite and can thus be orthogonally diagonalized as $\mathcal{L} = \boldsymbol{Q} \boldsymbol{\Lambda} \boldsymbol{Q}^T = \sum_{i=1}^{n} \lambda_i \boldsymbol{q}_i \boldsymbol{q}_i^T$, where $\boldsymbol{\Lambda} := \mathrm{diag}(\lambda_1, \ldots, \lambda_n)$ is a diagonal matrix with the eigenvalues on the main diagonal and $\boldsymbol{Q}$ is an orthogonal matrix containing the corresponding normalized eigenvectors $\boldsymbol{q}_1, \ldots, \boldsymbol{q}_n \in \mathbb{R}^n$ as its columns.

A detailed study (see, e.g., [18]) of the eigenvalues reveals that $0 = \lambda_1 \leqslant \lambda_2 \leqslant \cdots \leqslant \lambda_n \leqslant 2$. We can interpret the $\lambda_i, i \in [n]$ as the frequency magnitudes and the $\boldsymbol{q}_i$ as the corresponding Fourier modes. We accordingly define the *Fourier transform* of a signal vector $\boldsymbol{x} \in \mathbb{R}^n$ by $\hat{\boldsymbol{x}}[i] = \langle \boldsymbol{x}, \boldsymbol{q}_i \rangle$ for $i \in [n]$. The corresponding inverse Fourier transform is given by $\boldsymbol{x} = \sum_{i=1}^{n} \hat{\boldsymbol{x}}[i] \boldsymbol{q}_i$. Note that this can be written compactly as $\hat{\boldsymbol{x}} = \boldsymbol{Q}^T \boldsymbol{x}$ and $\boldsymbol{x} = \boldsymbol{Q} \hat{\boldsymbol{x}}$. Finally, we introduce the concept of *graph convolutions*. We define a filter $g : V \to \mathbb{R}$ defined on the set of nodes and want to convolve the corresponding filter vector $\boldsymbol{g} \in \mathbb{R}^n$ with a signal vector $\boldsymbol{x} \in \mathbb{R}^n$, i.e. $\boldsymbol{g} \star \boldsymbol{x}$. To explicitly compute this convolution, we recall that in the Euclidean setting, the convolution of two signals equals the product of their corresponding frequencies. This property generalizes to graphs [19] in the sense that $\widehat{(\boldsymbol{g} \star \boldsymbol{x})}[i] = \hat{\boldsymbol{g}}[i] \hat{\boldsymbol{x}}[i]$ for $i \in [n]$. Applying the inverse Fourier transform yields

$$\boldsymbol{g} \star \boldsymbol{x} = \sum_{i=1}^{n} \hat{\boldsymbol{g}}[i] \hat{\boldsymbol{x}}[i] \boldsymbol{q}_i = \sum_{i=1}^{n} \hat{\boldsymbol{g}}[i] \langle \boldsymbol{q}_i, \boldsymbol{x} \rangle \boldsymbol{q}_i = \boldsymbol{Q} \widehat{\boldsymbol{G}} \boldsymbol{Q}^T \boldsymbol{x},$$

where $\widehat{\boldsymbol{G}} := \mathrm{diag}(\hat{\boldsymbol{g}}) = \mathrm{diag}(\hat{\boldsymbol{g}}[1], \ldots, \hat{\boldsymbol{g}}[n])$. Hence, convolutional graph filters can be parameterized by considering the Fourier coefficients in $\widehat{\boldsymbol{G}}$.

Furthermore, it can be verified [20] that when these coefficients are defined as polynomials $\hat{\boldsymbol{g}}[i] := \sum_k \gamma_k \lambda_i^k$ for $i \in \mathbb{N}$ of the Laplacian eigenvalues in $\boldsymbol{\Lambda}$ (i.e. $\widehat{\boldsymbol{G}} = \sum_k \gamma_k \boldsymbol{\Lambda}^k$), the resulting filter convolution are localized in space and can be written in terms of $\mathcal{L}$ as $\boldsymbol{g} \star \boldsymbol{x} = \sum_k \gamma_k \mathcal{L}^k \boldsymbol{x}$ without requiring spectral decomposition of the normalized Laplacian. This motivates the standard practice [4, 20–22] of using filters that have polynomial forms, which we follow here as well.

For completeness, we note there exist alternative frameworks that generalize signal processing notions to graph domains, such as [23], which emphasizes the construction of complex filters that requires a notion of signal phase on graphs. However, extensive study of such alternatives is out of scope for the current work, which thus relies on the well-established (see, e.g., [24]) framework described here.

## 3 Graph Convolutional Network

Graph convolutional networks (GCNs), introduced in [4], consider semi-supervised settings where only a small potion of the nodes is labeled. They leverage intrinsic geometric information encoded in the adjacency matrix $\boldsymbol{W}$ together with node labels by constructing a convolutional filter parametrized by $\hat{\boldsymbol{g}}[i] := \theta(2 - \lambda_i)$, where the choice of a single learnable parameter is made to avoid overfitting. This parametrization yields a convolutional filtering operation given by

$$\boldsymbol{g}_\theta \star \boldsymbol{x} = \theta \left( \boldsymbol{I}_n + \boldsymbol{D}^{-1/2} \boldsymbol{W} \boldsymbol{D}^{-1/2} \right) \boldsymbol{x}. \tag{1}$$

The matrix $\boldsymbol{I}_n + \boldsymbol{D}^{-1/2} \boldsymbol{W} \boldsymbol{D}^{-1/2}$ has eigenvalues in $[0, 2]$. This could lead to vanishing or exploding gradients. This issue is addressed by the following renormalization trick [4]: $\boldsymbol{I}_n + \boldsymbol{D}^{-1/2} \boldsymbol{W} \boldsymbol{D}^{-1/2} \to \tilde{\boldsymbol{D}}^{-1/2} \tilde{\boldsymbol{W}} \tilde{\boldsymbol{D}}^{-1/2}$, where $\tilde{\boldsymbol{W}} := \boldsymbol{I}_n + \boldsymbol{W}$ and $\tilde{\boldsymbol{D}}$ a diagonal matrix with $\tilde{\boldsymbol{D}}[v_i, v_i] := \sum_{j=1}^{n} \tilde{\boldsymbol{W}}[v_i, v_j]$ for $i \in [n]$. This operation replaces the features of the nodes by a weighted average of itself and its neighbors. Note that the repeated execution of graph convolutions will enforce similarity throughout higher-order neighborhoods with order equal to the number of stacked layers. Setting $\boldsymbol{A} := \tilde{\boldsymbol{D}}^{-1/2} \tilde{\boldsymbol{W}} \tilde{\boldsymbol{D}}^{-1/2}$, the complete layer-wise propagation rule takes the form $\boldsymbol{h}_j^\ell = \sigma \left( \sum_{i=1}^{N_{\ell-1}} \theta_{ij}^\ell \boldsymbol{A} \boldsymbol{h}_i^{\ell-1} \right)$, where $\ell$ indicates the layer with $N_\ell$ neurons, $\boldsymbol{h}_j^\ell \in \mathbb{R}^n$ the activation vector of the $j^{th}$ neuron, $\theta_{ij}^\ell$ the learned parameter of the convolution with the $i^{th}$ incoming activation vector from the preceding layer and $\sigma(.)$ an element-wise applied activation function. Written in matrix notation, this gives

$$\boldsymbol{H}^\ell = \sigma \left( \boldsymbol{A} \boldsymbol{H}^{\ell-1} \boldsymbol{\Theta}^\ell \right), \tag{2}$$

where $\boldsymbol{\Theta}^\ell \in \mathbb{R}^{N_{\ell-1} \times N_\ell}$ is the weight-matrix of the $\ell^{th}$ layer and $\boldsymbol{H}^\ell \in \mathbb{R}^{n \times N_\ell}$ contains the activations outputted by the $\ell^{th}$ layer.

We remark that the above explained GCN model can be interpreted as a low-pass operation. For the sake of simplicity, let us consider the convolutional operation (Eq. 1) before the reparametrization trick. If we observe the convolution operation as the summation $\boldsymbol{g}_\theta \star \boldsymbol{x} = \sum_{i=1}^n \boldsymbol{\gamma}_i \hat{\boldsymbol{x}}[i] \boldsymbol{q}_i$, we clearly see that higher weights $\boldsymbol{\gamma}_i = \theta(2 - \lambda_i)$ are put on the low-frequency harmonics, while high-frequency harmonics are progressively less involved as $0 = \lambda_1 \leqslant \lambda_2 \leqslant \cdots \leqslant \lambda_n \leqslant 2$. This indicates that the model can only access a diminishing portion of the original information contained in the input signal the more graph convolutions are stacked. This observation is in line with the well-known oversmoothing problem [8] related to GCN models. The repeated application of graph convolutions will successively smooth the signals of the graph such that nodes cannot be distinguished anymore.

## 4 Geometric Scattering

In this section, we recall the construction of geometric scattering on graphs. This construction is based on the *lazy random walk* matrix

$$\boldsymbol{P} := \frac{1}{2}\big(\boldsymbol{I}_n + \boldsymbol{W}\boldsymbol{D}^{-1}\big),$$

which is closely related to the *graph random walk* defined as a Markov process with transition matrix $\boldsymbol{R} := \boldsymbol{W}\boldsymbol{D}^{-1}$. The matrix $\boldsymbol{P}$ however allows self loops while normalizing by a factor of two in order to retain a Markov process. Therefore, considering a distribution $\boldsymbol{\mu}_0 \in \mathbb{R}^n$ of the initial position of the lazy random walk, its positional distribution after $t$ steps is encoded by $\boldsymbol{\mu}_t = \boldsymbol{P}^t \boldsymbol{\mu}_0$.

As discussed in [12], the propagation of a graph signal vector $\boldsymbol{x} \in \mathbb{R}^n$ by $\boldsymbol{x}_t = \boldsymbol{P}^t \boldsymbol{x}$ performs a low-pass operation that preserves the zero-frequencies of the signal while suppressing high frequencies. In geometric scattering, this low-pass information is augmented by introducing the *wavelet* matrices $\boldsymbol{\Psi}_k \in \mathbb{R}^{n \times n}$ of scale $2^k$, $k \in \mathbb{N}_0$,

$$\begin{cases} \boldsymbol{\Psi}_0 := \boldsymbol{I}_n - \boldsymbol{P}, \\ \boldsymbol{\Psi}_k := \boldsymbol{P}^{2^{k-1}} - \boldsymbol{P}^{2^k} = \boldsymbol{P}^{2^{k-1}}\big(\boldsymbol{I}_n - \boldsymbol{P}^{2^{k-1}}\big), \quad k \geq 1. \end{cases} \tag{3}$$

This leverages the fact that high frequencies can be recovered with multiscale wavelet transforms, e.g., by decomposing nonzero frequencies into dyadic frequency bands. The operation $(\boldsymbol{\Psi}_k \boldsymbol{x})[v_i]$ collects signals from a neighborhood of order $2^k$, but extracts multiscale differences rather than averaging over them. The wavelets in Eq. 3 can be organized in a filter bank $\{\boldsymbol{\Psi}_k, \boldsymbol{\Phi}_K\}_{0 \leq k \leq K}$, where $\boldsymbol{\Phi}_K := \boldsymbol{P}^{2^K}$ is a pure low-pass filter. The telescoping sum of the matrices in this filter bank constitutes the identity matrix, thus enabling to reconstruct processed signals from their filter responses. Further studies of this construction and its properties (e.g., energy preservation) appear in [25] and related work.

*Geometric scattering* was originally introduced in the context of whole-graph classification and consisted of aggregating *scattering features*. These are stacked wavelet transforms (see Fig. 1) parameterized via tuples $p := (k_1, \ldots, k_m) \in \cup_{m \in \mathbb{N}} \mathbb{N}_0^m$ containing the bandwidth scale parameters, which are separated by element-wise absolute value nonlinearities[2] according to

$$\boldsymbol{U}_p \boldsymbol{x} := \boldsymbol{\Psi}_{k_m} |\boldsymbol{\Psi}_{k_{m-1}} \ldots |\boldsymbol{\Psi}_{k_2}|\boldsymbol{\Psi}_{k_1}\boldsymbol{x}||\ldots|, \tag{4}$$

where $m$ corresponds to the length of the tuple $p$. The scattering features are aggregated over the whole graph by taking $q^{th}$-order moments over the set of nodes,

$$\boldsymbol{S}_{p,q}\boldsymbol{x} := \sum_{i=1}^n |\boldsymbol{U}_p \boldsymbol{x}[v_i]|^q. \tag{5}$$

As our work is devoted to the study of node-based classification, we reinvent this approach in a new context, keeping the scattering transforms $\boldsymbol{U}_p$ on a node-level by dismissing the aggregation step in Eq. 5. For each tuple $p$, we define the following scattering propagation rule, which mirrors the GCN rule but replaces the low-pass filter by a geometric scattering operation resulting in

$$\boldsymbol{H}^\ell = \sigma\left(\boldsymbol{U}_p \boldsymbol{H}^{\ell-1} \boldsymbol{\Theta}^\ell\right). \tag{6}$$

We note that in practice, we only choose a subset of tuples, which is chosen as part of the network design explained in the following section.

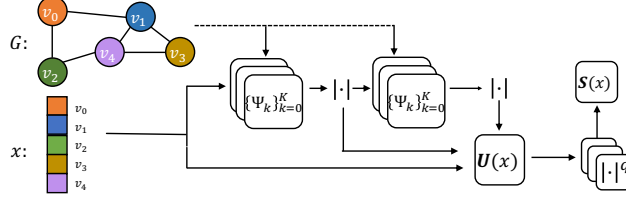

Figure 1: Illustration of geom. scattering at the node level ($U(x) = \{U_p x : p \in \mathbb{N}_0^m, m = 0, 1, 2\}$) and at the graph level ($S(x) = \{S_{p,q} x : q \in \mathbb{N}, p \in \mathbb{N}_0^m, m = 0, 1, 2\}$), extracted according to the wavelet cascade in Eqs. 3-5. While $m \leq 2$ orders are illustrated here, more can be used in general.

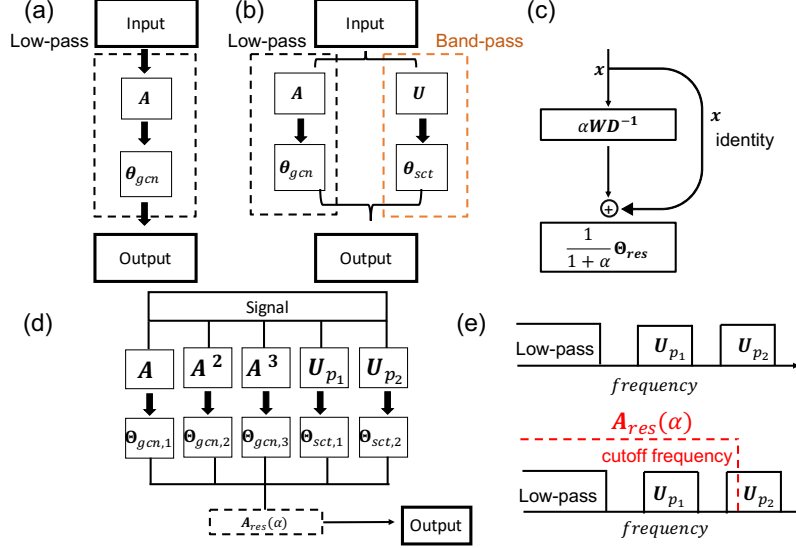

Figure 2: (a,b) Comparison between GCN and our network: we add band-pass channels to collect different frequency components; (c) Graph residual convolution layer; (d) Band-pass layers; (e) Schematic depiction in the frequency domain.

## 5 Combining GCN and Scattering Models

To combine the benefits of GCN models and geometric scattering adapted to the node level, we now propose a hybrid network architecture as shown in Fig. 2. It combines low-pass operations based on GCN models with band-pass operations based on geometric scattering. To define the layer-wise propagation rule, we introduce

$$H_{gcn}^{\ell} := \left[ H_{gcn,1}^{\ell} \parallel \ldots \parallel H_{gcn,C_{gcn}}^{\ell} \right] \quad \text{and} \quad H_{sct}^{\ell} := \left[ H_{sct,1}^{\ell} \parallel \ldots \parallel H_{sct,C_{sct}}^{\ell} \right],$$

which are the concatenations of channels $\left\{ H_{gcn,k}^{\ell} \right\}_{k=1}^{C_{gcn}}$ and $\left\{ H_{sct,k}^{\ell} \right\}_{k=1}^{C_{sct}}$, respectively. Every $H_{gcn,k}^{\ell}$ is defined according to Eq. 2 with the slight modification of added biases and powers of $A$,

$$H_{gcn,k}^{\ell} := \sigma \left( A^k H^{\ell-1} \Theta_{gcn,k}^{\ell} + B_{gcn,k}^{\ell} \right).$$

Note that every GCN filter uses a different propagation matrix $A^k$ and therefore aggregates information from $k$-step neighborhoods. Similarly, we proceed with $H_{sct,k}^{\ell}$ according to Eq. 6 and calculate

$$H_{sct,k}^{\ell} := \sigma \left( U_{p_k} H^{\ell-1} \Theta_{sct,k}^{\ell} + B_{sct,k}^{\ell} \right),$$

where $p_k \in \bigcup_{m \in \mathbb{N}} \mathbb{N}_0^m$, $k = 1, \ldots, C_{sct}$ enables scatterings of different orders and scales. Finally, the GCN components and scattering components get concatenated to

$$H^{\ell} := \left[ H_{gcn}^{\ell} \parallel H_{sct}^{\ell} \right]. \tag{7}$$

The learned parameters are the weight matrices $\boldsymbol{\Theta}^{\ell}_{gcn,k}, \boldsymbol{\Theta}^{\ell}_{sct,k} \in \mathbb{R}^{N_{\ell-1} \times N_{\ell}}$ coming from the convolutional and scattering layers. These are complemented by vectors of the biases $\boldsymbol{b}^{\ell}_{gcn,k}, \boldsymbol{b}^{\ell}_{sct,k} \in \mathbb{R}^{N_{\ell}}$, which are transposed and vertically concatenated $n$ times to the matrices $\boldsymbol{B}_{gcn,k}, \boldsymbol{B}_{sct,k} \in \mathbb{R}^{n \times N_{\ell}}$. To simplify notation, we assume here that all channels use the same number of neurons ($N_{\ell}$). Waiving this assumption would slightly complicate the notation but works perfectly fine in practice.

In this work, for simplicity, and because it is sufficient to establish our claim, we limit our architecture to three GCN channels and two scattering channels as illustrated in Fig. 2 (b). Inspired by the aggregation step in classical geometric scattering, we use $\sigma(.) := | \, . \, |^{q}$ as our nonlinearity. However, unlike the powers in Eq. 5, the $q^{th}$ power is applied at the node-level here instead of being aggregated as moments over the entire graph, thus retaining the distinction between node-wise activations.

We set the input of the first layer $\boldsymbol{H}^0$ to have the original node features as the graph signal. Each subchannel (GCN or scattering) transforms the original feature space to a new hidden space with the dimension determined by the number of neurons encoded in the columns of the corresponding submatrix of $\boldsymbol{H}^{\ell}$. These transformations are learned by the network via the weights and biases. Larger matrices $\boldsymbol{H}^{\ell}$ (i.e., more columns as the number of nodes in the graph is fixed) indicate that the weight matrices have more parameters to learn. Thus, the information in these channels can be propagated well and will be sufficiently represented.

In general, the width of a channel is relevant for the importance of the captured regularities. A wider channel suggests that these frequency components are more critical and need to be sufficiently learned. Reducing the width of the channel suppresses the magnitude of information that can be learned from a particular frequency window. For more details and analysis of specific design choices in our architecture we refer the reader to the ablation study provided in the supplement.

# 6 Graph Residual Convolution

Using the combination of GCN and scattering architectures, we collect multiscale information at the node level. This information is aggregated from different localized neighborhoods, which may exhibit vastly different frequency spectra. This comes for example from varying label rates in different graph substructures. In particular, very sparse graph sections can cause problems when the scattering features actually learn the difference between labeled and unlabeled nodes, creating high-frequency noise. In the classical geometric scattering used for whole-graph representation, geometric moments were used to aggregate the node-based information, serving at the same time as a low-pass filter. As we want to keep the information localized on the node level, we choose a different approach inspired by skip connections in residual neural networks [26]. Conceptually, this low-pass filter, which we call *graph residual convolution*, reduces the captured frequency spectrum up to a cutoff frequency as depicted in Fig. 2 (e).

The graph residual convolution matrix, governed by the hyperparameter $\alpha$, is given by $\boldsymbol{A}_{res}(\alpha) = \frac{1}{\alpha+1}(\boldsymbol{I}_n + \alpha \boldsymbol{W} \boldsymbol{D}^{-1})$ and we apply it after the hybrid layer of GCN and scattering filters. For $\alpha = 0$ we get the identity (no cutoff), while $\alpha \to \infty$ results in $\boldsymbol{R} = \boldsymbol{W} \boldsymbol{D}^{-1}$. This can be interpreted as an interpolation between the completely lazy (i.e., stationary) random walk and the non-resting (i.e., with no self-loops) random walk $\boldsymbol{R}$. We apply the graph residual layer on the output $\boldsymbol{H}^{\ell}$ of the Scattering GCN layer (Eq. 7). The update rule for this step, illustrated in Fig. 2 (c), is then expressed by $\boldsymbol{H}^{\ell+1} = \boldsymbol{A}_{res}(\alpha) \boldsymbol{H}^{\ell} \boldsymbol{\Theta}_{res} + \boldsymbol{B}_{res}$, where $\boldsymbol{\Theta}_{res} \in \mathbb{R}^{N \times N_{\ell+1}}$ are learned weights, $\boldsymbol{B}_{res} \in \mathbb{R}^{n \times N_{\ell+1}}$ are learned biases (similar to the notations used previously), and $N$ is the number of features of the concatenated layer $\boldsymbol{H}^{\ell}$. If $\boldsymbol{H}^{\ell+1}$ is the final layer, we choose $N_{\ell+1}$ equal to the number of classes.

# 7 Additional Information Introduced by Node-level Scattering Features

Before empirically verifying the viability of the proposed architecture in node classification tasks, we first discuss and demonstrate the additional information provided by scattering channels beyond that provided by traditional GCN channels. We first consider information carried by node features, treated as graph signals, and in particular their regularity over the graph. As discussed in Sec. 3, such regularity is traditionally considered only via smoothness of signals over the graph, as only low frequencies are retained by (local) smoothing operations. Band-pass filtering, on the other hand,

can retain other forms of regularity such as periodic or harmonic patterns. The following lemma demonstrates this difference between GCN and scattering channels.

**Lemma 1.** *Consider a cyclic graph on $2n$ nodes, $n \in \mathbb{N}$, and let $\boldsymbol{x} \in \mathbb{R}^{2n}$ be a 2-periodic signal on it (i.e., $\boldsymbol{x}_{2\ell-1} = a$ and $\boldsymbol{x}_{2\ell} = b$, for $\ell \in [n]$ for some $a \neq b \in \mathbb{R}$). Then, for any $\theta \in \mathbb{R}$, the GCN filtering $\boldsymbol{g}_\theta \star \boldsymbol{x}$ from Eq. 1 yields a constant signal, while the scattering filter $\boldsymbol{\Psi}_0 \boldsymbol{x}$ from Eq. 3 still produces a 2-periodic signal. Further, this result extends to any finite linear cascade of such filters (i.e., $\boldsymbol{g}_\theta \star \cdots \star \boldsymbol{g}_\theta \star \boldsymbol{x}$ or $\boldsymbol{\Psi}_0 \cdots \boldsymbol{\Psi}_0 \boldsymbol{x}$ with $k \in \mathbb{N}$ filter applications in each).*

While this is only a simple example, it already indicates a fundamental difference between the regularity patterns considered in graph convolutions compared to our approach. Indeed, it implies that if a smoothing convolutional filter encounters alternating signals on isolated cyclic substructures within a graph, their node features become indistinguishable, while scattering channels (with appropriate scales, weights and bias terms) will be able to make this distinction. Moreover, this difference can be generalized further beyond cyclic structures to consider features encoding two-coloring information on constant-degree bipartite graphs, as shown in the following lemma. We refer the reader to the supplement for a proof of this lemma, which also covers the previous one as a particular case, as well as numerical examples illustrating the results in these two lemmas.

**Lemma 2.** *Consider a bipartite graph on $n \in \mathbb{N}$ nodes with constant node degree $\beta$. Let $\boldsymbol{x} \in \mathbb{R}^n$ be a 2-coloring signal (i.e., with one part assigned constant $a$ and the other $b$, for some $a \neq b \in \mathbb{R}$). Then, for any $\theta \in \mathbb{R}$, the GCN filtering $\boldsymbol{g}_\theta \star \boldsymbol{x}$ from Eq. 1 yields a constant signal, while the scattering filter $\boldsymbol{\Psi}_0 \boldsymbol{x}$ from Eq. 3 still produces a (non-constant) 2-coloring of the graph. Further, this result extends to any finite linear cascade of such filters (i.e., $\boldsymbol{g}_\theta \star \cdots \star \boldsymbol{g}_\theta \star \boldsymbol{x}$ or $\boldsymbol{\Psi}_0 \cdots \boldsymbol{\Psi}_0 \boldsymbol{x}$ with $k \in \mathbb{N}$ filter applications in each).*

Beyond the information encoded in node features, graph wavelets encode geometric information even when it is not carried by input signals. Such a property has already been established, e.g., in the context of community detection, where white noise signals can be used in conjunction with graph wavelets to cluster nodes and reveal faithful community structures [27]. To demonstrate a similar property in the context of GCN and scattering channels, we give an example of a simple graph structure with two cyclic substructures of different sizes (or cycle lengths) that are connected by one bottleneck edge. In this case, it can be verified that even with constant input signals, some geometric information is encoded by its convolution with graph filters as illustrated in Fig. 3 (we refer the reader to the supplement for exact calculation of filter responses). However, as demonstrated in this case, while the information provided by the GCN filter responses $\boldsymbol{g}_\theta \star \boldsymbol{x}$ from Eq. 1 is not constant, it does not distinguish between the two cyclic structures (and a similar pattern can be verified for $\boldsymbol{A}\boldsymbol{x}$). Formally, each node $u$ in one cycle is shown to have at least one node $v$ in the other with the same filter response (i.e., $\boldsymbol{g}_\theta \star \boldsymbol{x}(u) = \boldsymbol{g}_\theta \star \boldsymbol{x}(v)$). In contrast, the information extracted by the wavelet filter response $\boldsymbol{\Psi}_3 \boldsymbol{x}$ (used in geometric scattering) distinguishes between cycles and would allow for their separation. We note that this property generalizes to other cycle lengths as discussed in the supplement, but leave more extensive study of geometric information encoding in graph wavelets to future work.

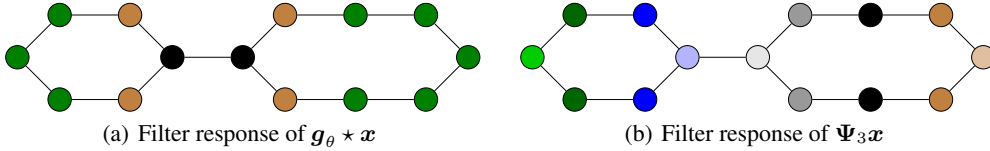

(a) Filter response of $\boldsymbol{g}_\theta \star \boldsymbol{x}$  (b) Filter response of $\boldsymbol{\Psi}_3 \boldsymbol{x}$

Figure 3: Filter responses for (a) the GCN filter (Eq. 1) and (b) a scattering filter applied to a constant signal $\boldsymbol{x}$ over a graph with two cyclic substructures connected by a single-edge bottleneck. Color coding differs slightly between plots, but is consistent within each plot, indicating nodes with numerically indistinguishable response values.

## 8   Empirical Results

To evaluate our Scattering GCN approach, we compare it to several established methods for semi-supervised node classification, including the original GCN [4], which is known to be subject to the

oversmoothing problem, as discussed in [8], and Sec. 1 and 3 here. Further, we compare our approach with two recent methods that address the oversmoothing problem. The approach in [8] directly addresses oversmoothing in GCNs by using partially absorbing random walks [28] to mitigate rapid mixing of node features in highly connected graph regions. The graph attention network (GAT) [10] indirectly addresses oversmoothing by training adaptive node-wise weighting of the smoothing operation via an attention mechanism. Furthermore, we also include two alternatives to GCN networks based on Chebyshev polynomial filters [20] and belief propagation of label information [29] computed via Gaussian random fields. Finally, we include two baseline approaches to verify the contribution of our hybrid approach compared to compared to the classifier from [13] that is solely based on handcrafted graph-scattering features, and compared to SVM classifier acting directly on node features without considering graph edges, which does not incorporate any geometric information.

The methods from [4, 8, 10, 20, 29] were all executed using the original implementations accompanying their publications. These are tuned and evaluated using the standard splits provided for the benchmark datasets for fair comparison. We ensure that the reported classification accuracies agree with previously published results when available. The tuning of our method (including hyperparameters and composition of GCN and scattering channels) on each dataset was done via grid search (over a fixed set of choices for all datasets) using the same cross validation setup used to tune competing methods. For further details, we refer the reader to the supplement, which contains an ablation study evaluating the importance of each component in our proposed architecture.

Our comparisons are based on four popular graph datasets with varying sizes and connectivity structures summarized in Tab. 1 (see, e.g., [30] for Citeseer, Cora, and Pubmed, and [17] for DBLP). We order the datasets by increasing connectivity structure, reflected by their node degrees and edges-to-nodes ratios. As

Table 1: Dataset characteristics: number of nodes, edges, and features; mean $\pm$ std. of node degrees; ratio of #edges to #nodes.

| Dataset | Nodes | Edges | Features | Degrees | $\frac{\text{Edges}}{\text{Nodes}}$ |
|---------|-------|-------|----------|---------|------------|
| Citeseer | 3,327 | 4,732 | 3,703 | 3.77$\pm$3.38 | 1.42 |
| Cora | 2,708 | 5,429 | 1,433 | 4.90$\pm$5.22 | 2.00 |
| Pubmed | 19,717 | 44,338 | 500 | 5.50$\pm$7.43 | 2.25 |
| DBLP | 17,716 | 52,867 | 1639 | 6.97$\pm$9.35 | 2.98 |

discussed in [8], increased connectivity leads to faster mixing of node features in GCN, exacerbating the oversmoothing problem (as nodes quickly become indistinguishable) and degrading classification performance. Therefore, we expect the impact of scattering channels and the relative improvement achieved by Scattering GCN to correspond to the increasing connectivity order of datasets in Tab. 1, which is maintained for our reported results in Tab. 2 and Fig. 4.

We first consider test classification accuracy reported in Tab. 2, which shows that our approach outperforms other methods on three out of the four considered datasets. On the remaining one (namely Citeseer) we are only outperformed by GAT. However, we note that this dataset has the weakest connectivity structure (see Tab. 1) and the most informative node features (e.g., achieving 61.1% accuracy via linear SVM without considering any graph information). In contrast, on

Table 2: Classification accuracy (top two marked in bold; best one underlined) of Scattering GCN on four benchmark datasets compared to four other GNNs [10, 8, 4, 20], a non-GNN approach [29] based on belief propagation, a pure graph scattering baseline [13], and a nongeometric baseline only using node features with linear SVM.

| Model | Citeseer | Cora | Pubmed | DBLP |
|-------|----------|------|--------|------|
| Scattering GCN (ours) | **71.7** | **84.2** | **79.4** | **81.5** |
| GAT [10] | **72.5** | **83.0** | 79.0 | 66.1 |
| Partially absorbing [8] | 71.2 | 81.7 | **79.2** | 56.9 |
| GCN [4] | 70.3 | 81.5 | 79.0 | 59.3 |
| Chebyshev [20] | 69.8 | 78.1 | 74.4 | 57.3 |
| Label Propagation [29] | 58.2 | 77.3 | 71.0 | 53.0 |
| Graph scattering [13] | 67.5 | 81.9 | 69.8 | **69.4** |
| Node features (SVM) | 61.1 | 58.0 | 49.9 | 48.2 |

DBLP, which has the richest connectivity structure and least informative features (only 48.2% SVM accuracy), we significantly outperform GAT (over 15% improvement), which itself significantly outperforms all other methods (by 6.8% or more) except for the graph scattering baseline from [13].

Next, we consider the impact of training size on classification performance as we are interested in semi-supervised settings where only a small portion of nodes in the graph are labelled. Fig. 4 (top) presents the classification accuracy (on validation set) for the training size reduced to 20%, 40%,

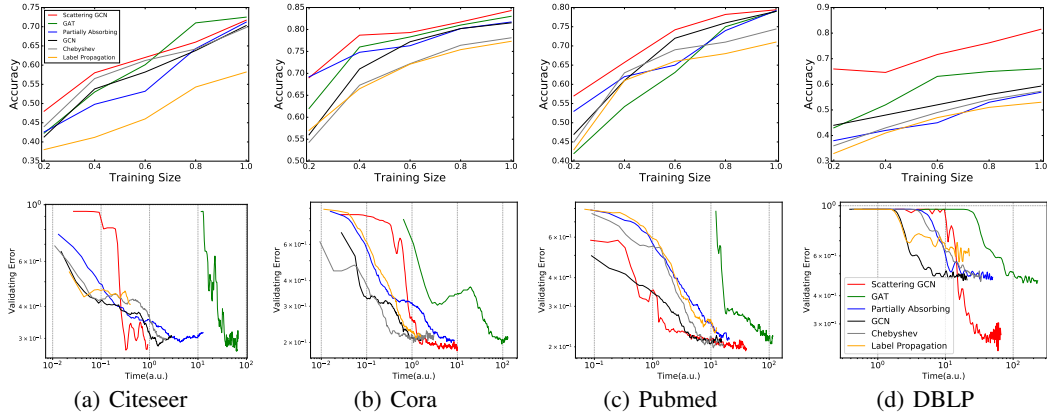

| (a) Citeseer | (b) Cora | (c) Pubmed | (d) DBLP |

Figure 4: Impact of training set size (top) and training time (bottom) on classification accuracy and error (correspondingly); training size measured relative to the original training size of each dataset; training time and validation error plotted in logarithmic scale; runtime measured for all methods on the same hardware, using original implementations accompanying their publications.

60%, 80% and 100% of the original training size available for each dataset. These results indicate that Scattering GCN generally exhibits greater stability to sparse training conditions compared to other methods. Importantly, we note that on Citeseer, while GAT outperforms our method for the original training size, its performance degrades rapidly when training size is reduced below 60% of the original one, at which point Scattering GCN outperforms all other methods. We also note that on Pubmed, even a small decrease in training size (e.g., 80% of original) creates a significant performance gap between Scattering GCN and GAT, which we believe is due to node features being less independently informative in this case (see baseline in Tab. 2) compared to Citeseer and Cora.

Finally, in Fig. 4 (bottom), we consider the evolution of (validation) classification error during the training process. Overall, our results indicate that the training of Scattering GCN reaches low validation errors significantly faster than Partially Absorbing and GAT[3], which are the two other leading methods (in terms of final test accuracy in Tab. 2). On Pubmed, which is the largest dataset considered here (by number of nodes), our error decays at a similar rate to that of GCN, showing a notable gap over all other methods. On DBLP, which has a similar number of nodes but significantly more edges, Scattering GCN takes longer to converge (compared to GCN), but as discussed before, it also demonstrates a significant (double-digit) performance lead compared to all other methods.

# 9   Conclusion

Our study of semi-supervised node-level classification tasks for graphs presents a new approach to address some of the main concerns and limitations of GCN models. We discuss and consider richer notions of regularity on graphs to expand the GCN approach, which solely relies on enforcing smoothness over graph neighborhoods. This is achieved by incorporating multiple frequency bands of graph signals, which are typically not leveraged in traditional GCN models. Our construction is inspired by geometric scattering, which has mainly been used for whole-graph classification so far. Our results demonstrate several benefits of incorporating the elements presented here (i.e., scattering channels and residual convolutions) into GCN architectures. Furthermore, we expect the incorporation of these elements together in more intricate architectures to provide new capabilities of pattern recognition and local information extraction in graphs. For example, attention mechanisms could be used to adaptively tune scattering configurations at the resolution of each node, rather than the global graph level used here. We leave the exploration of such research avenues for future work.

## Broader Impact

Node classification in graphs is an important task that gains increasing interest nowadays in multiple fields looking into network analysis applications. For example, they are of interest in social studies, where a natural application is the study of social networks and other interaction graphs. Other popular application fields include biochemistry and epidemiology. However, this work is computational in nature and addresses the foundations of graph processing and geometric deep learning. As such, by itself, it is not expected to raise ethical concerns nor to have adverse effects on society.

## Acknowledgments and Disclosure of Funding

The authors would like to thank Dongmian Zou for fruitful discussions. This work was partially funded by IVADO Professor startup & operational funds, IVADO Fundamental Research Project grant PRF-2019-3583139727, and NIH grant R01GM135929. The content provided here is solely the responsibility of the authors and does not necessarily represent the official views of the funding agencies.

## Footnotes

[2]In a slight deviation from previous work, here $\boldsymbol{U}_p$ does not include the outermost nonlinearity in the cascade.

[3]The horizontal shift shown for GAT in Fig. 4 (bottom), indicating increased training runtime (based on the original implementation accompanying [10]), could be explained by its optimization process requiring more weights than other methods and an intensive gradient computations driven not only graph nodes, but also by graph edges considered in the multihead attention mechanism.

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
