[Supplementary Material]

# Scattering GCN: Overcoming Oversmoothness in Graph Convolutional Networks – Supplement

**Yimeng Min**    **Frederik Wenkel**    **Guy Wolf**

## A    Proofs and Illustrative Examples of Lemmas 1 and 2

(a) Bipartite graph                    (b) Cyclic graph

Figure 4: Illustrative examples for Lemma 1 and 2 in Sec. 7 of the main paper.

*Proof of Lemma 1.* Note that $G_{cyc}^{2n}$ is a bipartite graph with constant node degree $\beta = 2$. Therefore, the proof of Lemma 1 can be seen as special case of Lemma 2, which is proved below. □

*Proof of Lemma 2.* We first notice that if $g_{\frac{1}{2}} \star x$ is constant, then $g_\theta \star x = 2\theta(g_{\frac{1}{2}} \star x)$ is constant for any $\theta$. Furthermore, for the considered class of graphs, $D = \beta I_n$ with $\beta > 0$, implying that $D^{-1} = \frac{1}{\beta} I_n$ and $D^{-1/2} = \frac{1}{\sqrt{\beta}} I_n$. Therefore, as a direct result of Eq. 1 in the main paper, it holds that

$$g_{\frac{1}{2}} \star x = \left( \frac{1}{2} I_n + \frac{1}{2\beta} W \right) x = Px. \tag{8}$$

Similarly, it is easily verified that any $k \in \mathbb{N}$ applications of the convolution with $g_\theta$ (for any $\theta \in \mathbb{R}$) can be written as $2^k \theta^k P^k x$. Furthermore, since $P$ is column-stochastic and (here) symmetric (thus also row-stochastic), we have $Pc = c$ for any constant signal $c = c\mathbf{1}_{2n}$. Thus, it is sufficient to show that $Px$ is a constant signal to verify the first claim of the lemma.

We consider $G_{bi,\beta}^n = (V, E)$. For any node $v \in V$, according to Eq. 8, we can write

$$(Px)[v] = \overbrace{P[v,v]}^{=\frac{1}{2}} x[v] + \sum_{u \in \mathcal{N}(v)} \overbrace{P[v,u]}^{=\frac{1}{2\beta}} x[u] + \sum_{w \in V^v} \overbrace{P[v,w]}^{=0} x[w],$$

where we denote by $\mathcal{N}(v)$ the neighborhood of the node $v$ and set $V^v := V \setminus (\{v\} \cup \mathcal{N}(v))$. This implies that

$$(\boldsymbol{P}\boldsymbol{x})[v] = \frac{\boldsymbol{x}[v]}{2} + \sum_{u \in \mathcal{N}(v)} \frac{\boldsymbol{x}[u]}{2\beta}. \tag{9}$$

We now consider a 2-coloring signal $\boldsymbol{x} \in \mathbb{R}^n$. W.l.o.g., let $\boldsymbol{x}[v] = a$, which implies $\boldsymbol{x}[u] = b$ for all $u \in \mathcal{N}(v)$. Now, since $|\mathcal{N}(v)| = \beta$, it holds

$$(\boldsymbol{P}\boldsymbol{x})[v] = \frac{a+b}{2},$$

thus verifying the first claim of the lemma as the choice of $v$ was arbitrary. Finally, it is now straightforward to verify the second claim as well, since the operation $\boldsymbol{\Psi}_0 \boldsymbol{x} = (\boldsymbol{I}_n - \boldsymbol{P})\boldsymbol{x} = \boldsymbol{x} - \frac{a+b}{2}\mathbf{1}_n$ retains a 2-coloring signal (the original colors are shifted by a constant: $-\frac{a+b}{2}$).  $\square$

## B   Geometric Information Encoded by Graph Wavelets (supp. info. Sec. 7)

Let $\mathcal{G}_{cyc}^{\geq \ell} := \{G_{cyc}^k : k \geq \ell\}$, $3 \leq \ell \in \mathbb{N}$, be the class of unweighted cyclic graphs of length greater than or equal to $\ell$. Furthermore, let $\mathcal{G}_\ell^\star$ be the class of graphs constructed by taking $n \geq 2$ cyclic graphs $G_1, \ldots, G_n$, arranging them in the order of the indexes, and connecting subsequent cycles with bottleneck edges as described in the following.

1. We take $G_1, G_n \in \mathcal{G}_{cyc}^{\geq 4}$ and $G_2, \ldots, G_{n-1} \in \mathcal{G}_{cyc}^{\geq \ell}$.

2. The (sub)graph $G_i$, $1 \leq i \leq n-1$, is connected by exactly one bottleneck edge to $G_{i+1}$.

3. No edge is connecting (sub)graphs $G_i$ and $G_j$ if $|i - j| \geq 2$.

4. For each $G_i$, $2 \leq i \leq n-1$, there are exactly two nodes in $G_i$ with bottleneck edges coming out of them, and these nodes are the farthest from each other in the cycle [1] (in shortest-path distance).

This construction essentially generalizes the graph demonstrated in Fig. 3 of the main paper (see Sec. 7). The following lemma shows that on such graphs, the filter responses of $\boldsymbol{g}_\theta$ for a constant signal will encode some geometric information, but will not distinguish between the cycles in the graph. Note that this result can also be generalized further to a chain that is closed by connecting $G_n$ and $G_1$ with a bottleneck edge if we further assume that $G_1, G_n \in \mathcal{G}_{cyc}^{\geq 7}$.

**Lemma 3.** *Let $G = (V, E)$ a graph of the class $\mathcal{G}_7^\star$. We consider a constant signal $\boldsymbol{c} = c\mathbf{1}_{|V|}$, for some $c \in \mathbb{R}$. Then, for all nodes $v \in V$ and for any $\theta \in \mathbb{R}$, the filter response $(\boldsymbol{g}_\theta \star \boldsymbol{c})[v]$ shares its value with at least one node of each other cyclic substructure.*

*Proof.* First, note that $V$ contains only the following two kinds of nodes. We refer to a node of degree 3 (those contained in a bottleneck edge) as a *hub*, while using the term *pass* for all other nodes (those of degree 2). Furthermore, due to the minimal cycle length, and the shortest-path distance requirement between hub nodes in the same cycle, only three types of neighborhoods can be encountered. Indeed, it is easy to see that each hub node has one hub neighbor and two pass neighbors, while pass nodes can either have two pass neighbors or one pass and one hub.

Next, we notice that for any $\theta$ and any $c$, the filter response $(\boldsymbol{g}_\theta \star \boldsymbol{c})[v] = c\theta(\boldsymbol{g}_1 \star \mathbf{1}_{|V|})[v]$ is fully determined by the neighborhood type of $v \in V$. Therefore, computing these boils down to a simple proof by cases:

1. Let $v$ be a hub, then the corresponding response is $(\boldsymbol{g}_\theta \star \boldsymbol{c})[v] = c\,\theta\left(1 + \frac{1}{3} + \frac{1}{\sqrt{6}} + \frac{1}{\sqrt{6}}\right) \approx 2.150 \cdot c\,\theta$.

2. Let $v$ be a pass with two pass neighbors, then the corresponding response is $(\boldsymbol{g}_\theta \star \boldsymbol{c})[v] = c\,\theta\left(1 + \frac{1}{2} + \frac{1}{2}\right) = 2 \cdot c\,\theta$.

3. Let $v$ be a pass with one hub neighbor and one pass neighbor, then the corresponding response is $(\boldsymbol{g}_\theta \star \boldsymbol{c})[v] = c\,\theta\left(1 + \frac{1}{\sqrt{6}} + \frac{1}{2}\right) \approx 1.908 \cdot c\,\theta$.

Finally, it is trivial to see that every cyclic substructure from the class $\mathcal{G}_7^\star$ contains at least one hub and two passes connected to that hub. The existence of a pass only connected to other passes follows from the choice of the minimal cycle lengths together with the requirement that two hubs within a cycle have maximal distance from each other. □

(a) Filter response of $\boldsymbol{g}_\theta \star \boldsymbol{x}$      (b) Filter response of $\boldsymbol{\Psi}_3 \boldsymbol{x}$

Figure 5: Filter responses used in (a) GCN and (b) Scattering channels when applied to a constant signal $\boldsymbol{x}$ over a graph with two cyclic substuctures connected by a single edge bottleneck.

Let us now revisit the example given in the main paper (see Fig. 3 or a copy in Fig. 5). We consider a graph consisting of 14 nodes organized in 2 cycles ($v_1 \sim v_2 \cdots v_6 \sim v_1$ and $v_7 \sim v_8 \cdots v_{14} \sim v_7$ here) of different length (i.e., 6 and 8 here), which are connected with one single edge between any two nodes taken from different cycles ($v_4$ and $v_7$ here). As this is a specific case of Lemma 3, the filter responses of the GCN filter $\boldsymbol{g}_\theta$ for a constant signal $\boldsymbol{x}$ would indeed not distinguish between cycles as discussed in Sec. 7 of the main paper (with the pattern shown in Figs. 3 and 5 for $\theta = 1$). On the other hand, the filter responses of $\boldsymbol{\Psi}_3 \boldsymbol{x}$ on a constant signal (e.g., $\boldsymbol{x} = \mathbf{1}_{14} \in \mathbb{R}^{14}$) on this graph can be verified empirically as follows:

|  | $v_1$ | $v_2$ | $v_3$ | $v_4$ | $v_5$ | $v_6$ |
|---|---|---|---|---|---|---|
| $\boldsymbol{\Psi}_3 \boldsymbol{x}$ ($10^{-3} \times$) | 33.0 | 20.3 | -10.7 | -52.3 | -10.7 | 20.3 |

|  | $v_7$ | $v_8$ | $v_9$ | $v_{10}$ | $v_{11}$ | $v_{12}$ | $v_{13}$ | $v_{14}$ |
|---|---|---|---|---|---|---|---|---|
| $\boldsymbol{\Psi}_3 \boldsymbol{x}$ ($10^{-3} \times$) | -53.5 | -13.9 | 11.2 | 19.8 | 19.4 | 19.8 | 11.2 | -13.9 |

These responses with appropriate color coding give the illustration in Fig. 2 in the main paper. A similar distinction between cycles with band-pass filters can also be empirically verified for other cases covered by Lemma 3. We leave further theoretical studies of this property of graph wavelets to future work.

## C   Further Discussion of Related Work

As many applied fields such as Bioinformatics and Neuroscience heavily rely on the analysis of graph-structured data, the study of reliable classification methods has received much attention lately. In this work, we focus on the particular class of semi-supervised classification tasks, where GCN models [1, 2] recently proved to be effective. Their theoretical studies reveal however that graph convolutions can be interpreted as Laplacian smoothing operations, which poses fundamental limitations on the approach. Another branch of GNNs, manifested in [3], introduces self-attention mechanisms to determine adequate node-wise neighborhoods, which in turn alleviate the mentioned shortcomings of GCN approaches. Further, [4] developed a theoretical framework based on graph signal processing, relying on the relation between frequency and feature noise, to show that GNNs perform a low-pass filtering on the feature vectors. In [5], multiple powers of the adjacency matrix were used to learn the higher-order neighborhood information, while [6] used Lanczos algorithm to construct a low-rank approximation of the graph Laplacian that efficiently gathers multiscale information, demonstrated on citation networks and the QM8 quantum chemistry dataset. Finally, [7] studied wavelets on graphs and collected higher-order neighborhood information based on wavelet transformation.

Together with the study of learned networks, recent studies have also introduced the construction of geometric scattering transforms, relying on manually crafted families of graph wavelet transforms [8–10]. Similar to the initial motivation of geometric deep learning to generalize convolutional neural networks, the geometric scattering framework generalizes the construction of Euclidean scattering

from [11] to the graph setting. Theoretical studies [e.g., 8, 12, 13] established energy preservation properties and the stability of these generalized scattering transforms to perturbations and deformations of graphs and signals on them. Moreover, the practical application of geometric scattering to whole-graph data analysis was studied in [9], achieving strong classification results on social networks and biochemistry data, which established the effectiveness of this approach.

As discussed in the main paper, this work aims to combine the complementary strengths of GCN models and geometric scattering and to provide a new avenue for incorporating richer notions of regularity in GNNs. Further, our construction integrates trained task-driven components in geometric scattering architectures. Finally, while most previous work on geometric scattering focused on whole-graph settings, we consider node-level processing, which requires new considerations about the construction.

# D    Technical Details

Similar to other neural networks, the presented Scattering GCN poses several architecture choices and hyperparameters that can be tuned and affect its performance. For simplicity, we set the last layer before the output classification to be the residual convolution layer and only consider one or two hybrid layers before it, each consisting of three GCN channels and two scattering channels. We note that this restricted setup simplifies the network tuning process and was sufficient in our experiments to obtain promising results (outperforming other methods, as shown in the main paper), but can naturally be generalized further to deeper or wider architectures in practice. Furthermore, based on preliminary results, *Cora*, *Citeseer* and *Pubmed* were set to use only one hybrid layer as the addition of a second one was not cost-effective (considering the added complexity of a grid search for tuning hyperparameters based on validation results). For *DBLP*, two layers were used due to a significant increase in performance. We note that even with a single hybrid layer our model achieves $73.1\%$ test accuracy (compared to the reported $81.5\%$ for two layers) and still significantly outperforms GAT ($66.1\%$) and the other methods (below $60\%$).

**Validation & testing procedure:**    All tests were done using train-validation-test splits of the datasets, where validation accuracy is used for tuning hyperparameters and test accuracy is reported in the comparison table. The same splits were used for all methods for a fair comparison. To ensure our evaluation is comparable with previous work, for *Citeseer*, *Cora* and *Pubmed* we used the same settings as in [1], following the standard practice used in other work reporting results on these datasets. For *DBLP*, as far as we know, no common standard is established in the literature. Here, we used a ratio of $5:1:1$ between train, validation, and test.

**Hyperparameter tuning:**    Given the general network architectures and train-validation-test splits, the hyperparameter tuning was performed for each dataset using grid search guided by validation accuracy. The grid covered the tuning of the redsidual convolution via $\alpha$, the nonlinarity exponent $q$ (inspired by scattering moments), the scattering channel configuration (i.e., scales used in these two channels), and the widths of channels in the network. The results of this tuning process are presented in the following table.

| | $\alpha$ | $q$ | Scat. config.: | | Channel widths: | | | | |
| --- | --- | --- | --- | --- | --- | --- | --- | --- | --- |
| | | | $U_{J_1}$ | $U_{J_2}$ | $A^1$ | $A^2$ | $A^3$ | $U_{p_1}$ | $U_{p_2}$ |
| *Citeseer* | 0.50 | 4 | $\Psi_2$ | $\Psi_2\vert\Psi_3\vert$ | 10 | 10 | 10 | 9 | 30 |
| *Cora* | 0.35 | 4 | $\Psi_1$ | $\Psi_3$ | 10 | 10 | 10 | 11 | 6 |
| *Pubmed* | 1.00 | 4 | $\Psi_1$ | $\Psi_2$ | 10 | 10 | 10 | 13 | 14 |
| *DBLP (1st layer)* | 1.00 | 4 | $\Psi_1$ | $\Psi_2$ | 10 | 10 | 10 | 30 | 30 |
| *DBLP (2nd layer)* | 0.10 | 1 | $\Psi_1$ | $\Psi_2$ | 40 | 20 | 20 | 20 | 20 |

It should be noted that for *DBLP*, the hybrid-layer parameters are shared between the two used layers in order to simplify the tuning process, which was generally less exhaustive than for the other three datasets, since even with limited tuning our method significantly outperformed all other methods. That being said, we note that the difference in effectiveness of architecture and hyperparameter choices (as well as the increased performance of our approach compared to others) observed in this case could be a result of the significantly different connectivity exhibited by its graph as discussed

briefly in Sec. 8 (of the main paper). Regardless, as more exhaustive tuning would not degrade (and likely improve) the results obtained from Scattering GCN, we view our limited tuning done here as sufficient for establishing the advantages provided by our approach over other methods and leave a more intensive study of the *DBLP* dataset to future work.

**Hardware & software environment:** All comparisons were executed on the same HPC cluster with intel i7-6850K CPU and NVIDIA TITAN X Pascal GPU. Scattering GCN was implemented in Python using the PyTorch [14] framework. Implementations of all other methods were taken directly from the code accompanying their publications.

# E Ablation Study

The two main components of our Scattering GCN architecture contribute together to achieve significant improvements over pure GCN models. Namely, these are the additional scattering channels (i.e., $U_{J_1}$ and $U_{J_2}$) and the residual convolution (controlled by the hyperparemeter $\alpha$). To further explore their contribution and the hyperparameter space for their tuning, Tab. 3-7 show classification results over the Cora dataset for $\alpha = 0.01, 0.1, 0.35, 0.5, 1.0$ (controlling the residual convolution layer) over multiple scattering channel configurations. For presentation brevity and simplicity, we focus our presented ablation benchmark on this dataset here, but note that similar results are also observed on the other datasets. The rows and columns in each table denote the two scattering channels used in the Scattering GCN, together with the three GCN channels (i.e., for $A^k$, $k = 1, 2, 3$).

First, we consider the importance of the residual graph convolution layer. We note that setting $\alpha = 0$ effectively ignores this layer (i.e., by setting its operation to be the identity), while increasing $\alpha$ makes the filtering provided by it to be more dominant until $\alpha = 1$, where it essentially becomes a random walk-based low-pass filter. Therefore, to evaluate the importance of this component of our architecture, it is sufficient to evaluate the impact of $\alpha$ on the classification accuracy. Indeed, our results (see Tab. 3-7) indicate that increasing $\alpha$ to non-negligible nonzero values improves classification performance, which we interpret to be due to the removal of high-frequency noise. However, when $\alpha$ further increases (in particular when $\alpha = 1$ in this case) the smoothing provided by this layer degrades the performance to a level close to the traditional GCN [1]. Therefore, these results suggest that when well tuned (e.g., as done via grid search in this work), the graph residual

Table 3: Classification accuracies on Cora with $\alpha = 0.01$ with average accuracy 80.4% over all scales.

| ACCURACY | $\Psi_1$ | $\Psi_2$ | $\Psi_3$ | $\Psi_2|\Psi_3|$ | $\Psi_1|\Psi_2|$ |
|---|---|---|---|---|---|
| $\Psi_1$ | 0.808 | 0.808 | 0.805 | 0.806 | 0.806 |
| $\Psi_2$ | 0.809 | 0.809 | 0.806 | 0.806 | 0.806 |
| $\Psi_3$ | 0.802 | 0.804 | 0.801 | 0.801 | 0.800 |
| $\Psi_2|\Psi_3|$ | 0.802 | 0.804 | 0.801 | 0.800 | 0.799 |
| $\Psi_1|\Psi_2|$ | 0.802 | 0.804 | 0.801 | 0.800 | 0.800 |

Table 4: Classification accuracies on Cora with $\alpha = 0.1$ with average accuracy 80.9% over all scales.

| ACCURACY | $\Psi_1$ | $\Psi_2$ | $\Psi_3$ | $\Psi_2|\Psi_3|$ | $\Psi_1|\Psi_2|$ |
|---|---|---|---|---|---|
| $\Psi_1$ | 0.813 | 0.813 | 0.812 | 0.808 | 0.809 |
| $\Psi_2$ | 0.817 | 0.817 | 0.810 | 0.810 | 0.815 |
| $\Psi_3$ | 0.810 | 0.808 | 0.801 | 0.800 | 0.806 |
| $\Psi_2|\Psi_3|$ | 0.812 | 0.811 | 0.801 | 0.800 | 0.809 |
| $\Psi_1|\Psi_2|$ | 0.813 | 0.811 | 0.802 | 0.802 | 0.809 |

Table 5: Classification accuracies on Cora with $\alpha = 0.35$ with average accuracy 83.5%.

| ACCURACY | $\Psi_1$ | $\Psi_2$ | $\Psi_3$ | $\Psi_2|\Psi_3|$ | $\Psi_1|\Psi_2|$ |
|---|---|---|---|---|---|
| $\Psi_1$ | 0.838 | 0.836 | 0.835 | 0.837 | 0.837 |
| $\Psi_2$ | **0.842** | 0.836 | 0.835 | 0.838 | 0.837 |
| $\Psi_3$ | 0.835 | 0.836 | 0.833 | 0.833 | 0.832 |
| $\Psi_2|\Psi_3|$ | 0.835 | 0.836 | 0.833 | 0.833 | 0.831 |
| $\Psi_1|\Psi_2|$ | 0.835 | 0.836 | 0.832 | 0.833 | 0.831 |

Table 6: Classification accuracies on Cora with $\alpha = 0.5$ with average accuracy 82.7% over all scales.

| ACCURACY | $\Psi_1$ | $\Psi_2$ | $\Psi_3$ | $\Psi_2|\Psi_3|$ | $\Psi_1|\Psi_2|$ |
|---|---|---|---|---|---|
| $\Psi_1$ | 0.828 | 0.828 | 0.836 | 0.836 | 0.835 |
| $\Psi_2$ | 0.828 | 0.833 | 0.830 | 0.830 | 0.827 |
| $\Psi_3$ | 0.821 | 0.829 | 0.826 | 0.826 | 0.826 |
| $\Psi_2|\Psi_3|$ | 0.820 | 0.828 | 0.826 | 0.825 | 0.825 |
| $\Psi_1|\Psi_2|$ | 0.821 | 0.829 | 0.824 | 0.824 | 0.824 |

Table 7: Classification accuracies on Cora with $\alpha = 1.0$ with average accuracy 82.3% over all scales.

| ACCURACY | $\Psi_1$ | $\Psi_2$ | $\Psi_3$ | $\Psi_2|\Psi_3|$ | $\Psi_1|\Psi_2|$ |
|---|---|---|---|---|---|
| $\Psi_1$ | 0.817 | 0.818 | 0.827 | 0.827 | 0.824 |
| $\Psi_2$ | 0.820 | 0.819 | 0.824 | 0.824 | 0.823 |
| $\Psi_3$ | 0.826 | 0.823 | 0.823 | 0.823 | 0.821 |
| $\Psi_2|\Psi_3|$ | 0.825 | 0.823 | 0.823 | 0.823 | 0.821 |
| $\Psi_1|\Psi_2|$ | 0.822 | 0.823 | 0.823 | 0.823 | 0.821 |

convolution plays a critical role in improving results, which can also be seen by the results of the hyperparameter tuning shown in the previous section.

Next, we consider the scales used in the two scattering channels of our hybrid architecture, which correspond to the rows and columns of Tab. 3-7 here. While our results show that the network is relatively robust to this choice, we can observe that generally utilizing purely second-order coefficients gives slightly worse results than either first-order ones or a mix of first- and second-order coefficients. Nevertheless, most configurations of scattering scales (with appropriate choice of $\alpha$) give better results than pure GCN, thus indicating that added information is extracted by scattering channels.

We note that for $\alpha = 0.35$ (Tab. 5), all scale configurations outperform GAT (83%) and all other reported methods in Tab. 2 (of the main paper). As a result, even the average accuracy over all scale configurations (83.5%) in this case shows an improvement over these other methods, thus further establishing the advantage of our approach. It is important to mention that while this improvement is affected by the tuning of the residual convolution, it also relies on the addition of scattering channels. Indeed, by itself, the residual convolution layer only applies a low-pass (smoothing) filter and therefore, without scattering channels, would essentially be equivalent to a conventional GCN.

We remark that the results presented in this work are based on a limited grid search, while the ablation study here indicates that many of the possible configurations provide noticeable classification performance improvement over other methods. It is likely that the reported results, in fact, provide a lower bound on the improvement attained by Scattering GCN, while a more exhaustive optimization of the architecture and its hyperparameters can further improve and solidify its advantages. We leave such exhaustive study for future work, which will also consider the incorporation of other advanced components (e.g., attention mechanisms) in the model architecture.

Finally, to further validate the importance of band-pass information added by the presented architecture here, we provide an ablation study of the impact each channel has on classification performance. We focus here on $\alpha = 0.35$ with the best configuration on Cora (i.e., achieving 84.2% accuracy when all channels are used). Then, we remove each of the band-pass or low-pass channels individually from the network, and reevaluate network performance with the remaining four channels. Our results, presented in Tab. 8, indicate that while information captured by $A, A^2$ and $A^3$ is important for the classification task, which is to be expected given the prevalence of such filters in GCNs, the band-pass information extracted by the scattering channels (with $\Psi_2$ and $\Psi_1$ in this case) plays a crucial role in achieving the performance of our method. In particular, we note that $\Psi_1$ in this case has a major impact on the accuracy, driving the difference between underperforming and outperforming GAT, thus strengthening the claim that important information in graph features is contained in higher frequencies extracted by band-pass filtering, which is not recovered by smoothing operations.

Table 8: Impact of removing each individual channel from the optimal configuration on Cora, while classifying using the remaining four channels. Full Scattering GCN accuracy provided for reference.

| REMOVED CHANNEL | $A$ | $A^2$ | $A^3$ | $\Psi_2$ | $\Psi_1$ | SCATTERING GCN |
|---|---|---|---|---|---|---|
| ACCURACY | 82.0 | 80.7 | 80.9 | 83.7 | 82.7 | 84.2 |

## F   Implementation

Python code accompanying this work is available on github.com/dms-net/scatteringGCN.

## Footnotes

[1] For cycles of odd length, the choice of the node to connect is ambiguous (as there are two qualifying nodes), but the claim holds for either choice