[Reviews · NeurIPS 2020]

Review 1

Summary and Contributions: The authors introduce a hybrid method based on Geometric Scattering and Graph Convolutional Networks. They introduce a Graph residual convolution approach to aggregate multi-scale information and describe some of the properties of the introduced architecture.

Strengths: It's a novel approach relevant to recent literature and probably a step in the right direction when it comes to graph classification. The results are good. The paper is written clearly.

Weaknesses: The Geometric Scattering approach is a rather ad-hoc choice for the task and properties that it introduces, e.g. section 7, are not experimentally demonstrated.

Correctness: The methods are correct. To prove all claims more experiments should be necessary.

Clarity: Yes it is very clear

Relation to Prior Work: Yes prior work is clearly discussed.

Reproducibility: Yes

Additional Feedback: The additional information sketched by the two lemmas in section 7 are not verified empirically in section 8. The authors claim that introducing wavelets improves sensitivity in high frequencies. Has that been verified experimentally? Scattering Transforms typically average over high frequencies (to achieve stability) and are known to lose high-frequency information. It is positive that the authors are attempting a hybrid model but the rationale of using scattering for high frequencies can be challenged and experimental evidence for the contrary are weak. Have you considered comparing with [1] which loosens the constraints on wavelet structure? Can you include performance of graph scattering methods in Tables 1-2. The results as they are make it difficult to draw any conclusion with respect to the quality of the contribution, i.e. what brings the performance? There is little effort to verify the rationale. Minor comments: line 111 small potion/portion [1] https://arxiv.org/pdf/2006.05722.pdf Post rebuttal feedback: On high frequencies, scattering compared to DL representations considerably fails at capturing high frequencies but I guess that would lead us to futile debate. If you do what you say in the rest of the rebuttal I believe the quality of the paper will improve and therefore I maintain my acceptance score.


Review 2

Summary and Contributions: *** Post rebuttal update *** I have read the authors' response and am satisfied with way they addressed the points raised and the modifications they'll introduce based on my feedback; thus, my 'accept' score remains unchanged. ********************************************** The paper proposes a modification of the graph convolutional network (GCN) architecture to add the scattering transform (Mallat, 2012) and a residual layer. The scattering transform layer has a geometric interpretation (analogous to multiscale structure in graph signal processing) and is intended to deal with the oversmoothness problem in GCNs. The residual layer is intended to keep information localized at a node level. The paper provides an algorithmic modification to GCN and a detailed empirical study to support the superior performance of the architecture on semi supervised graph labeling classification tasks.

Strengths: The contribution is mostly methodological; the paper also connects ideas from signal processing on graphs and neural networks in a new and interesting way. The paper is solidly grounded in the theory behind scattering transforms and the broader signal processing on graphs work of the last few years. The paper is relevant to the NeurIPS community as it provides an architecture that can enable practitioners working on semi supervised graph classification tasks.

Weaknesses: While the application of the scattering transform is explained from an intuitive point of view, the paper can be made stronger by fleshing out a few remaining issues: -- R is lazy random walk (line 224). The statement "which can be interpreted as an interpolation between the completely lazy random walk and the non-resting random walk R" is ambiguous and needs to be clarified. -- How is the residual layer helping in this architecture and why? The scattering layer already 'collects' information at the node level as defined (low- vs band pass filtering). Why do we need the extra threshold on the wavelets in Fig 1e?

Correctness: I did not find inaccurate claims, although there are some points that can be further fleshed out as highlighted in the "Weaknesses" section above.

Clarity: Overall the paper is clearly written and barring a few typos here and there it is of very high quality in terms of clarity.

Relation to Prior Work: The paper discusses prior work (scattering transforms and relevant GCN architectures) in much details and sufficiently well describes how it differs from related work.

Reproducibility: Yes

Additional Feedback:


Review 3

Summary and Contributions: This paper introduces an extension and combination of two existing methods (graph convolution and geometric scattering) and applies it to a novel application (node classification). The proposed combination alleviate undesirable oversmoothing of previous methods which is particularly important in node classification (whereas previous methods were focusing on whole graph classification problems for which that over-smoothing was less of an issue). The paper reintroduces most of the necessary tools and existing method, explains their limitations (over-smoothing) to motivate the proposed approach. Then, the proposed method is described. Its non-over-smoothing behavior is convincingly demonstrated on toy-task (periodic and colored graph) as well as on experimental classification task.

Strengths: Straightforward to read, apart perhaps some notations issue (see additional feedback). Good motivation and efficient and elegant solution to address the issue.

Weaknesses: Perhaps some lack of clarity on the notations. I found the notations where often misleading and inconsistent with prior work on scattering, which makes the paper a bit hard to read in my opinion. Although, perhaps on the novelty side, in section 5 the proposed approach seems to be a concatenation of prior work GCN together some additional high frequencies features, and it seems a super-set of a representation is bound to do a bit better than the original representation.

Correctness: The paper seems correct overall, maybe with a reservation on some of the notations and equations that I found a bit confusing. The experimental methodology seems quite nice. For prior work methods, they downloaded original code and reran it on the data they used, making sure the number were matching original publications when available, which I think is really the best possible practice, although I haven't actually checked those numbers.

Clarity: The motivations and the general explanations are nice and straightforward. However, some of the notations and equations were a bit unclear to me, see additional feedback for details. I might probably be willing to increase my rating if the equations were easier to follow and the relation to prior work notations were a bit more straightforward. The paper is a bit too packed maybe, for example Figure 1 has a lot of different purpose, Table 1 and 2 are a bit too close to the rest of the text etc...

Relation to Prior Work: There seems to be a solid prior work discussion both in the motivations and in the experimental section. Again, really nice practice of downloading and rerunning prior work code and making sure the numbers agree with past publications.

Reproducibility: Yes

Additional Feedback: l49-50: "compute with attention mechanisms computed from": double use of computed l111: "potion": portion? l121: "h_j^l = ...": j doesn't appear in the LHS? This is a bit confusing. If I understand correctly, actually A and \tilde D actually both depend on j through their def l117? Then, maybe it would be clearer to also index those A and \tilde D by j and theta by i j or am I missing something? l133 potion again l157: "J:= (k_1, k_m)", I found this notation quite confusing as J is meant to be the maximum / smoothing scale in other scattering papers, whereas here it is the tuple of frequencies. Also Phi in scattering paper is usually used for the last smoothing operator, whereas here it is used for the unsmoothed version? l161: "we reinvent": reapply? l189: I don't think defining B from b adds much to the readability.


Review 4

Summary and Contributions: The paper introduces scattering transforms to tackle the oversmoothing behavior of standard graph convolutional neural nets. Additionally, graph residual convolutions are introduced as a mechanism to restrict the captured frequency spectrum. The proposed approach is evaluated on standard benchmarks for semi-supervised node classification in citation graphs, and comparisons are made with state-of-the-art geometric deep learning methods and other baselines.

Strengths: The paper identifies an innovative solution to an important limitation of GCNs. The combination of classical GCNs with band-pass channels through scattering transforms seems very promising. Another strength is the self-contained nature of the paper with a pretty complete yet concise introduction of the necessary background theory. The experiment on reducing the amount of labelled nodes is very informative and relevant for practical applications.

Weaknesses: The main weakness in my opinion is the experimental evaluation which is limited to the well known citation graph benchmarks. Although, these provide a good way to compare with published methods and additional baselines, it is unclear how well these results translate to more interesting applications that are discussed in the introduction and used to motivate the work. Another weakness is the relatively little insights provided about why and how exactly the scattering transform helps with the node classification. I appreciate Sec. 7 which does a theoretical analysis of the complementary nature of the extracted information, but it would also be valuable to have more practical insights. For example, from the experiments it is quite clear that the proposed method shines for the DBLP benchmark where a lot of the information is hidden in the connectivity structure of the graph. But what exactly is it that the scattering layers extract from there what the other methods can't. At least to me this isn't very clear.

Correctness: Seems correct, but I could not follow all derivations without digging deeper into the signal processing theory (which is not my main area of expertise).

Clarity: The paper is dense but generally well written. The background sections on graph signal processing and GCNs are very helpful.

Relation to Prior Work: I believe relevant work is cited and appropriate comparisons are made to the relevant state-of-the-art approaches.

Reproducibility: Yes

Additional Feedback: I'd be interested to hear whether the proposed approach could also benefit from an attention mechanism similar to GAT. I wasn't entirely sure about the setup for the experiment where the training size is reduced. Is this taking a fixed graph and then simply hiding an increasing portion of the node labels, or is the graph structure different between the settings with reduced training size? Are the number of nodes for which labels are predicted the same between each setting? Is each unlabelled node always connected to at least one labelled node or does the reduction of training size also mean that the nearest labelled node might be further away in the low training size regime? A schematic illustration of the geometric scattering may help with the accessibility of the paper. A toy example with a simple node classification problem might be used for that (similar to Fig. 2). Typo in l. 326 "interesred" -> "interested" The broader impact statement is inadequate. The authors statement that "This work is computational in nature and addresses the foundations of graph processing and geometric deep learning. As such, by itself, it is not expected to raise ethical concerns nor to have adverse effects on society." is missing the point of the impact statement. Graph neural networks are heavily used for the analysis of social networks (e.g., fake news detection) and thus take center stage when it comes to ethical concerns and there are huge societal implications. To clarify, this has not impacted my review score but though it should be mentioned here. ---- AFTER REBUTTAL: I am happy with the rebuttal which answered the questions I had (mostly about possible extensions and experimental setup). I think this is a good paper and I am keeping my recommendation as is.

[Author Response · NeurIPS 2020]

**Author Response - Submission 7862 (NeurIPS 2020):** We thank the reviewers for their valuable comments and for recommending acceptance. Reviewer #1 says *"it's a novel approach"* and *"a step in the right direction"*. Reviewer #2 says it *"connects ideas [...] in a new and interesting way"* and *"is solidly grounded in the theory behind scattering transforms"*. Reviewer #3 says it has *"good motivation and efficient and elegant solution"* and *"the experimental methodology seems quite nice"*. Reviewer #4 says it *"identifies an innovative solution to an important limitation of GCNs"* and mentions *"the experiment on reducing the amount of labelled nodes is very informative"*. All typos and minor comments will be fixed in the final version, and the addition of a 9th page will be used to address comments about the paper being a bit packed (albeit well written). Further questions and concerns are addressed in the following:

**Reviewer #1:** *"the additional information sketched by the two lemmas in section 7 are not verified empirically"* **Reply:** We will add concrete examples of (colored) cyclic and bipartite graphs to demonstrate and verify the properties shown in these lemmas. ♦ *"Scattering Transforms typically average over high frequencies (to achieve stability) and are known to lose high-frequency information."* **Reply:** Respectfully, this statement is inaccurate. One of the main strengths of traditional scattering is its ability to capture and aggregate high frequency information thanks to the demodulation provided by the complex modulus (while losing the phase). This has been demonstrated in multiple applications involving image and audio textures, audio source separation, and inverse problems (all critically depending on high frequency information). The analogy in the graph case is not perfect (e.g., due to ill-defined demodulation), but we will add further discussion clarifying this point in the final version. ♦ *"Have you considered comparing with [1]"* **Reply:** The arxiv version of [1] was first posted on June 10 (then published in ICML in July), and thus not available before the NeurIPS submission deadline. Nevertheless, we will add discussion referring to its alternative GFT and (relaxed) wavelet construction. However, direct comparison is not applicable here, since it does not provide node-level features, but rather graph-level ones, while considering classification of multiple signals over a single graph, rather than nodes on it. ♦ *"Can you include performance of graph scattering methods in Tables 1-2."* **Reply:** To our knowledge, Zou & Lerman (ACHA 2019) is the only previous graph scattering work reporting node classification (others focus solely on graph-level tasks). Their paper only reports $81.9\%$ on Cora, below both GAT and our method, but their GitHub code achieves $69.4\%$ on DBLP (better than GAT, but still significantly below our method) and underperforms even GCN on Citeseer ($67.5\%$) and PubMed ($69.8\%$). We will include these results in the tables, as requested. Moreover, we will extend the ablation study in the supplement to further emphasize the impact of bandpass information in scattering channels compared to lowpass GCN ones. It should be noted that since the addition of bandpass information is the main distinction between our method and others in Sec. 8, the impact of such information is evidently nonnegligible.

**Reviewer #2:** *"[line 224] is ambiguous and needs to be clarified"* **Reply:** We will revise this sentence to clarify the "completely lazy" random walk is the stationary one (only having self-loops) represented by the identity, while the nonresting one $R$ contains no self-loops. ♦ *"How is the residual layer helping in this architecture and why? [...] Why do we need the extra threshold on the wavelets in Fig 1e?"* **Reply:** The motivation for this layer is discussed in Sec. 6 and its impact is verified empirically in the ablation study in the supplement. Briefly, the wavelet cascade in scattering extracts high frequencies and may capture undesirable features (e.g., distinguishing labeled and unlabeled nodes). The residual convolution alleviates such artifacts while maintaining the localization of node features. We note that traditional scattering transforms also typically use a final smoothing step (e.g., via lowpass or moments), as pointed out also by R1.

**Reviewer #3:** We thank the reviewer for the detailed feedback on the notations. We will make sure to clarify every mentioned instance. ♦ *"l121: '$h_j^l = ...$': j doesn't appear in the [R]HS?"* **Reply:** The reviewer is correct - there was a missing index $j$ on $\theta_{ij}^\ell$, which are the individual weights in the matrix $\Theta^\ell$ (see Eq. 2, lines 125-126). We will fix this missing index in all instances including the one noticed by the reviewer and two others in lines 122-123. ♦ *"[...] notations [w]ere often misleading and inconsistent with prior work on scattering"* **Reply:** We note that we largely tried to follow prior geometric scattering notations (e.g., Gao et al. 2019), while bridging certain discrepancies with notations used in GCN work. That being said, we acknowledge the reviewer's comment that our use of $\Phi$ and $J$ may have been somewhat misleading. We will therefore rename $\Phi$ to $U$, which is sometimes used for the wavelet cascade (albeit here without the outermost nonlinearity that is added later in lines 179-180 – we will clarify this slight abuse of notation in a footnote), and rename $J$ to $p$, following the notation from Mallat 2012 (Definition 2.2, page 10 there) for the scattering pathways. We will also make an effort to verify no other notation conflicts remain in the paper, which have not been explicitly pointed out by the reviewers, and we will revise the notations to resolve them as necessary.

**Reviewer #4:** *"I'd be interested to hear whether the proposed approach could also benefit from an attention mechanism similar to GAT."* **Reply:** While out of scope for this work, this is an excellent point, and indeed we are currently in the process of developing a scattering attention network that combines these ideas showing promising preliminary results. ♦ *"[details regarding the] setup for the experiment where the training size is reduced"* **Reply:** In our experiments, we decrease the number of labeled nodes (i.e., "hiding" node labels) for training on the fixed graph while the nodes for validation and test are the same. In a low training size regime, reducing the training size means that the nearest labeled node can be further away. ♦ *"A schematic illustration of the geometric scattering may help with the accessibility"* **Reply:** We agree and will add one, similar to Fig. 2 from Gao et al. 2019 or Fig. 1 from Gama et al. NeurIPS 2019.

[Meta-Review · NeurIPS 2020]

This paper works on graph-based semi-supervised learning (more specifically node classification) and proposes to combine a geometric scattering transform into GCNs to overcome the over-smoothing issue of state-of-the-art GCNs for node classification. It also proposes graph residual convolutions to better aggregate the node information. The clarity, novelty, and significance are clearly above the bar of NeurIPS. The authors also did a good job in their rebuttal to address some concerns raised by reviewers. Thus, all of us have agreed to accept this paper for publication!